# Inhibition of Microglial Activation by Amitriptyline and Doxepin in Interferon-β Pre-Treated Astrocyte–Microglia Co-Culture Model of Inflammation [note 1]

**DOI:** 10.3390/brainsci13030493

**Published:** 2023-03-15

**Authors:** Timo Jendrik Faustmann, Marisa Wawrzyniak, Pedro M. Faustmann, Franco Corvace, Fatme Seval Ismail

**Affiliations:** 1Department of Psychiatry and Psychotherapy, Medical Faculty, Heinrich Heine University, 40225 Düsseldorf, Germany; timo.faustmann@uni-duesseldorf.de; 2Department of Neuroanatomy and Molecular Brain Research, Ruhr University Bochum, 44801 Bochum, Germany; marisa.wawrzyniak@ruhr-uni-bochum.de (M.W.); pedro.faustmann@rub.de (P.M.F.); franco.corvace@ruhr-uni-bochum.de (F.C.); 3Department of Neurology, University Hospital Knappschaftskrankenhaus Bochum, Ruhr University Bochum, 44892 Bochum, Germany

**Keywords:** interferon-β, doxepin, amitriptyline, inflammation, depression, astrocyte–microglia co-culture model

## Abstract

Depression may occur in patients with multiple sclerosis, especially during interferon-β (IFN-β) treatment, and therapy with antidepressants may be necessary. Interactions of IFN-β with antidepressants concerning glia-mediated inflammation have not yet been studied. Primary rat co-cultures of astrocytes containing 5% (M5, consistent with “physiological” conditions) or 30% (M30, consistent with “pathological, inflammatory” conditions) of microglia were incubated with 10 ng/mL amitriptyline or doxepin for 2 h, or with 2000 U/mL IFN-β for 22 h. To investigate the effects of antidepressants on IFN-β treatment, amitriptyline or doxepin was added to IFN-β pre-treated co-cultures. An MTT (3-(4,5-dimethylthiazol-2-yl)-2,5-diphenyltetrazolium bromide) assay was performed to measure the glial cell viability, immunocytochemistry was performed to evaluate the microglial activation state, and ELISA was performed to measure pro-inflammatory TNF-α and IL-6 cytokine concentrations. Incubation of inflammatory astrocyte–microglia co-cultures with amitriptyline, doxepin or IFN-β alone, or co-incubation of IFN-β pre-treated co-cultures with both antidepressants, significantly reduced the extent of inflammation, with the inhibition of microglial activation. TNF-α and IL-6 levels were not affected. Accordingly, the two antidepressants did not interfere with the anti-inflammatory effect of IFN-β on astrocytes and microglia. Furthermore, no cytotoxic effects on glial cells were observed. This is the first in vitro study offering novel perspectives in IFN-β treatment and accompanying depression regarding glia.

## 1. Introduction

Multiple sclerosis (MS) is a chronic inflammatory and autoimmune-mediated, demyelinating disease of the central nervous system (CNS), with a complex etiopathology including autoreactive T-cells, B-cells and antibodies, but also activated microglia and further CNS factors, leading to severe neuropsychiatric dysfunction [1]. Different forms of MS can be distinguished: relapsing–remitting MS (RRMS), secondary progressive MS (SPMS) and primary progressive MS (PPMS) [2]. There is strong evidence that glial cells such as astrocytes and microglia are involved in the pathophysiology of MS [1,2,3,4]. Microglial cells are the main CNS-resident innate immune cells and comprise 5 to 20% of the glial cells in the healthy brain [5]. Resting microglia can be distinguished morphologically by a small cell body and high ramification. Microglia respond to inflammation and other neuropathological diseases with transformation of the microglial phenotype from resting to activated [5]. Activated microglia are characterized by a large cellular diameter, rare short processes and several cytoplasmic vacuoles. Microglial cells seem to be involved in MS, with a higher number of activated microglia as well as pro-inflammatory cytokines [6]. Moreover, elevated levels of pro-inflammatory cytokines were linked to higher disease activity [7] and microglia activation was associated with increased demyelination in MS [8]. Astrocytes are the main glial cell population, performing different functions, such as building the blood–brain barrier, stabilizing neuronal networks and maintaining brain homeostasis. They are involved in MS pathogenesis in many ways [3,9,10].

Different therapeutic and immunomodulatory approaches for MS are known, e.g., interferon-β treatment (IFN-β) for RRMS and clinically isolated syndromes. The mechanism of action of IFN-β involves multiple effects, e.g., it activates the Janus kinase (JAK) signal transducers and activators of transcription (STAT) pathway, leading to activation of the innate and adaptive immune response, e.g., by increasing the number of antigen-presenting cells and CD56 natural killer cells [11,12,13,14]. Further, it enhances the expression and release of anti-inflammatory agents, while downregulating the expression of pro-inflammatory cytokines, and reduces permeability of the blood–brain barrier for inflammatory cells [11,12,13,14]. Exogenous IFN-β can influence the microglial immune response, but in terms of inflammatory processes, IFN-β can be produced by microglia themselves [15].

Importantly, both MS itself and the treatment of MS, e.g., with disease-modifying drugs such as IFN-β, can influence the patient’s mental health [16]. The lifetime prevalence for clinically significant depression in MS patients is approximately 50% [17,18,19]. Patients with a history of depression have a higher risk of developing depression under IFN-β treatment [20]. The interaction between immune, endocrine and neuronal pathways is assumed to contribute to IFN-related depression [21]. Possible biochemical/biological mechanisms include (1) IFN-therapy (α, β)-induced hyperactivity by the hypothalamo–pituitary–adrenal axis, releasing corticotrophin-releasing hormone (CRH), which increases adrenocorticotropic hormone and hence adrenal corticosterone release, and decreases serotonin and noradrenaline; (2) modulation of mood and sleep behavior by the activation of pro-inflammatory cytokines by IFN-α (e.g., interleukin-6 (IL-6), interleukin-1 (IL-1), tumor necrosis factor-α (TNF-α), caspase-4 and caspase-8); (3) µ-opioid receptor activation by IFN (α, β), leading to an increase in brain prostaglandin E2 and different regulation of enzymes (e.g., indolamine 2,3-dioxygenase (IDO), kynureninase), influencing the N-methyl-D-aspartate (NMDA) receptor, which results finally in excitotoxicity [21]. Therefore, therapy with antidepressants may be necessary in patients receiving IFN-β treatment. Otherwise, there are data indicating no correlation between IFN-β and depressive symptoms in MS [22].

The pathophysiology of depression remains largely unclear, but it has been postulated that inflammation significantly contributes to the development of depression. Pro-inflammatory cytokines such as IL-6 and TNF-α were elevated in patients with depression [23]. Aside from elevated cytokine levels, studies have also found an association between depression and microglia activation within the frontolimbic system [24,25]. Recently, it has been described that psychiatric comorbidities in MS (depression and anxiety) predict pain intensity and pain affect [26]. From this perspective, pharmacological and psychological treatment is effective in reducing depression in MS [27].

Two main pharmacological approaches in depression, which are approved by the European Medicines Agency (EMA) and the United States Food and Drug Administration (FDA), are doxepin and amitriptyline. Both are tricyclic antidepressants. Amitriptyline inhibits serotonin and norepinephrine re-uptake, but also further effects, e.g., on histaminergic and muscarinic receptors, are known [28]. Additionally, amitriptyline is used for the treatment of neuropathic pain, migraine and tension headache [29]. Doxepin inhibits the re-uptake of serotonin and norepinephrine, with further inhibitory dopaminergic and muscarinic effects [30,31]. Doxepin is also approved for the treatment of insomnia [30,31].

Both antidepressants are involved in glial interactions and inflammation, including the binding affinity of doxepin to astrocytes [32], the amitriptyline-mediated upregulation of microglial IBA-1 activation marker in a neuropathic pain model [33], a decrease in neuropathic pain, as well as the downregulation of IL-6 mRNA and interleukin-1β (IL-1β) mRNA mediated by both antidepressants [34]. Further, amitriptyline reduced the release of IL-1β and TNF-α from rat mixed glial and microglial cultures [35], reduced astrogliosis and showed anti-inflammatory effects (reduction in IL-1β protein levels) in a model of multiple system atrophy [36] and in a lipopolysaccharide (LPS)-induced inflammatory model of depression (reduction in IL-1β, IL-6, TNF-α, IFN-γ) [37].

An in vitro astrocyte–microglia co-culture model of inflammation by Faustmann et al., 2003 has been proven as a powerful tool to study the pharmacological effects on neuroinflammation [5,38,39,40,41], e.g., different anti-epileptic, immunomodulatory (e.g., monomethyl and dimethyl fumarate) and psychotropic (e.g., venlafaxine) drugs have already been studied [38,39,40,41,42,43,44,45,46,47]. In this co-culture model, a microglia fraction of 5–10% with a predominantly resting phenotype was related to the physiological state (M5), whereas the pathological, inflammatory state was characterized by an increased microglia fraction of 30–40% (M30) and activated phenotype [5,42]. Microglial activation was detected after the incubation of M5 co-cultures with the pro-inflammatory cytokines TNF-α, IL-1β, IL-6 and IFN-γ, whereas the incubation of inflammatory M30 co-cultures with the anti-inflammatory cytokine transforming growth factor (TGF)-β1 led to a reduction in microglial activation [42]. IFN-β was able to prevent the effects of the pro-inflammatory cytokines TNF-α, IL-1β and IFN-γ on M5 co-cultures [42].

Based on previous studies, the exact link between inflammation related to glial cells and the interactions of amitriptyline and doxepin with IFN-β treatment remains unclear, but could point to new directions and approaches in clinical neuropsychiatry.

In this study, we aimed to investigate the effects of both antidepressants doxepin and amitriptyline on IFN-β pre-treated astrocyte–microglia co-cultures. We hypothesized that amitriptyline and doxepin did not attenuate the anti-inflammatory effect of IFN-β treatment with regard to glia-mediated inflammation. For this purpose, the glial viability, microglial activation and morphology and cytokine release were examined in different set-ups of an in vitro astrocyte–microglia co-culture model of inflammation after incubation with amitriptyline, doxepin or IFN-β alone. To answer the question of whether antidepressants interfere with the IFN-β effects, co-cultures pre-treated with IFN-β were subsequently incubated with doxepin or amitriptyline.

## 2. Materials and Methods

### 2.1. Cell Culture

The preparation of astrocyte–microglia co-cultures from postnatal Wistar rats (P0-P2) was performed according to a protocol established by Faustmann et al., 2003 [5]. After approval by the local authorities in Bochum, Germany, all experiments were performed according to the German Animal Welfare Act and the ethical standards of the Ruhr University Bochum. Under sterile conditions, the postnatal rats were decapitated and their cerebellum was removed. After removal of the meninges and choroid plexus, the cerebral hemispheres were temporarily stored in ice-cooled phosphate-buffered saline (PBS) (containing 1.38 M NaCl, 27 mM KCl, 81 mM NaH_2_PO_4_, 14.7 mM K_2_H_2_PO_4_ (J.T. Baker, Deventer, The Netherlands)) until further processing [5,37,38,39,40]. The brains were homogenized using 0.1% trypsin (PAA laboratories, Pasching, Austria) for 30 min at 37 °C. The homogenate was then centrifuged at 500× *g* for 12 min. The pellet was resuspended in DNase I solution (Serva Electrophoresis, Heidelberg, Germany) (100 µL/mL with Dulbecco’s minimal essential medium (DMEM), Invitrogen, Karlsruhe, Germany) and incubated for 5 min at room temperature. Following this, washing and seeding steps of astrocyte–microglia co-cultures were performed according to the standardized protocol [5,38,39,40,41]. After approximately 5 days, the co-culture reached 100% confluence. Depending on mechanical influences such as the extent of shaking, a total of two subpopulations of astrocyte–microglia co-cultures were generated, containing either 5–10% (M5, representing “physiological” conditions) or 30–40% (M30, representing “pathological, inflammatory” conditions) of microglia [5,38,39,40,41].

### 2.2. Treatment of Cultures

Based on previous studies and corresponding to the measured CSF concentrations of treated patients with amitriptyline, doxepin and IFN-β, the primary rat astrocyte–microglia co-cultures (M5 or M30 condition) were incubated with 10 ng/mL amitriptyline (Sigma-Aldrich, Schnelldorf, Germany) or doxepin (Sigma-Aldrich, Schnelldorf, Germany) for 2 h, or with 2000 U/mL IFN-β (R&D Systems, Minneapolis, MN, USA) for 22 h [42,48,49]. Previously, it has been reported that plasma doxepin levels correlate with CSF levels [50]. Further, 10 h after drug administration of doxepin, only relatively low concentrations of its active metabolite were measured in CSF [48]. Similar to this, findings indicated that plasma amitriptyline levels correlate with CSF levels. Levels of amitriptyline in CSF were found to be around 9.3 ng/mL with a 250 mg/day oral dose on day 20 after the start of treatment [49]. Previously, experiments with our astrocyte–microglia co-culture model revealed a significant protective effect against pro-inflammatory cytokines (TNF-α, IL-1β, and IFN-γ) upon a previous co-incubation with IFN-β (all together 24 h). In the last 2 h of incubation (after 22 h of pre-incubation with IFN-β), the pro-inflammatory cytokines were added to the cultures [42]. According to this, to investigate the effects of antidepressants on IFN-β, cell cultures were incubated with IFN-β (2000 U/mL) for 24 h and co-incubated with amitriptyline (10 ng/mL) or doxepin (10 ng/mL) during the last 2 h of IFN-β incubation. The drugs were dissolved in 0.9% NaCl and diluted in the culture medium. Untreated co-cultures were used as controls.

### 2.3. MTT (3-(4,5-Dimethylthiazol-2-yl)-2,5-Diphenyltetrazolium Bromide) Assay

To analyze cell viability, proliferation and cytotoxicity, an MTT assay was performed (Roche Applied Sciences, Penzberg, Germany). For this purpose, 10,000 cells per well were seeded on a poly-L-lysine (PLL)-coated 96-well plate. Upon reaching 80% confluence, the cells were incubated with the drugs as described above. The assay itself was performed according to the Roche protocol and was not modified (Roche Applied Sciences, Penzberg, Germany). On the following day, the optical density was determined using a Bio-Rad microplate reader (München, Germany) at a wavelength of 560 nm.

### 2.4. Immunocytochemistry

Both astrocyte and microglial cell numbers as well as microglial phenotypes were determined by immunocytochemistry according to a standardized protocol [5,38,39,40,41]. The astrocyte–microglia co-cultures were seeded on PLL-coated coverslips at a cell count of 70,000 cells/coverslip in a 24-well plate. After incubation of the cells with the drugs as described above, they were fixed using 100% ice-cooled ethanol for 10 min and then washed three times with PBS. To avoid non-specific binding, cells were subsequently blocked with PBS-blocking solution containing 1% bovine serum albumin (BSA) and 10% horse serum (HS) (PAA Laboratories, Linz, Austria). After removal of the blocking solution, the primary antibody (mouse anti-ED1 (Serotec, Düsseldorf, Germany)) diluted in PBS-blocking solution (1:250) was added and incubated overnight at 4 °C. The next day, the astrocyte–microglia co-cultures were washed three times with PBS containing 1% BSA and then incubated with the secondary antibody (goat anti-mouse IgG conjugates (Alexa fluor^®^ 568) (Invitrogen, Karlsruhe, Germany)) diluted in PBS-blocking (1:500) solution for 1 h in the dark. For quantification of the cell numbers, immunocytochemically labeled cells were counterstained with DAPI (4,6-diamidino-2-phenyl-indol) (1:2500) (Invitrogen, Karlsruhe, Germany). By comparison of the number of ED1-stained microglia with the total number of DAPI-labeled cells, the ratio of microglia to astrocytes was detected. The evaluation was performed using a fluorescence microscope at 630× magnification. For statistical analyses, at least three fields of view per coverslip were counted and averaged. Furthermore, the phenotype of microglia could be determined based on the ED1-staining [5]. This allowed the subdivision of the astrocyte–microglia co-cultures into M5 (5–11% microglia) and M30 (>30% microglia). The resting ramified (RRT) microglia phenotype is characterized by small cell bodies (5–10 µm) with a small perinuclear, cytoplasmic rim and thin branching processes longer than the diameter of the cell body (Figure 1A). Typical features of the activated rounded phagocytic (RPT) phenotype are a large cellular diameter, rare short processes and several cytoplasmic vacuoles (Figure 1B). The intermediate (INT) phenotype can be distinguished by some thick pseudopodia longer than the diameter of the cell body and a perinuclear cytoplasmic rim with a few vesicles and vacuoles (Figure 1B) [5,38,39,40,41,47].

### 2.5. Nitric Oxide (NO) Assay

To determine the NO concentration, the measurement of nitrite (NO_2_), as a primary breakdown product of NO, was performed using a Griess NO assay (Promega, Mannheim, Germany) according to the manufacturer’s instructions. First, astrocyte–microglia co-cultures were seeded with a cell number of 10,000 cells per well on PLL-coated 96-well plate. Upon reaching 80% confluence, the cells were incubated with the drugs as described above. After addition of 50 µL sulfanilamide solution and incubation for 10 min in the dark, 50 µL NED (N-1-napthylethylenediamine dihydrochloride) solution was added and incubated again for 10 min. The absorbance was subsequently determined at 550 nm using a Bio-Rad microplate reader (Hercules, CA, USA).

### 2.6. Enzyme-Linked Immunosorbent Assay (ELISA)

Two kits for the measurement of supernatant concentrations of IL-6 and TNF-α were used to perform ELISA according to the manufacturer’s instructions (Promega, Madison, WI, USA; R&D Systems, Minneapolis, MN, USA). First, cells on PLL-coated 96-well plates were treated with 100 µL of the capture antibody (anti-rat TNF-α or anti-rat IL-6 diluted in PBS) and incubated overnight at 4 °C. After washing and blocking steps according to the protocol, 100 µL of the sample or standard was added. After incubation of the samples for 2 h, three washing steps were performed and 100 µL of detection antibody (biotinylated goat anti-rat IL-6 (400 ng/mL) or TNF-α (225 ng/mL)) was added for 2 h. After incubation with 100 μL streptavidin horseradish peroxidase (HRP) in the dark for 20 min, a washing step was performed. Next, the samples were incubated with 100 μL of Substrate Solution for 20 min. Finally, 50 μL of Stop Solution (2N H_2_SO_4_) was added. For the evaluation of the ELISA, the absorbance of the plate was determined at 450 nm (Bio-Rad Microplate Reader (Bio-Rad 550, Hercules, CA, USA)) and compared with the standard curve. This allowed the subsequent determination of the cytokine concentration.

### 2.7. Data Analyses and Statistics

GraphPad Prism version 7.0 for Windows (GraphPad Software, San Diego, CA, USA) and Microsoft Office 2010 were used for the statistical analysis of the MTT, NO assay and immunocytochemistry data. The Four Parameter Logistic Regression protocol from myassays.com (https://www.myassays.com; accessed on 1 December 2021) was used for the analysis and statistics of the ELISA results. Subsequent statistical analysis was again performed using GraphPad 7.0. The Kolmogorov–Smirnov test and D’Agostino–Pearson omnibus test were performed to analyze the normality of the data distribution. When normality was given, parametric tests were applied. Statistical significance was determined at *p* < 0.05. The results were presented as mean ± standard error of the mean.

## 3. Results

### 3.1. Effects of Doxepin, Amitriptyline and IFN-β on Glial Cell Viability

Incubation with doxepin, amitriptyline or IFN-β did not significantly change the glial cell viability in M5 co-cultures, representing “physiological” conditions (Figure 2A), and in M30 co-cultures, representing “pathological” conditions (Figure 2B) (*n* = 9). In addition, no significant changes in the glial cell viability were observed after co-incubation with doxepin or amitriptyline of IFN-β pre-treated co-cultures (*n* = 9) (Figure 2A,B) (*p* = ns).

### 3.2. Influence of Doxepin, Amitriptyline and IFN-β on Cell Numbers in Different Set-Ups of Astrocyte–Microglia Co-Culture Model of Inflammation

The incubation of physiological M5 co-cultures with 2000 U/mL IFN-β for 22 h resulted in a significant increase in the total cell number by 22.2% (*n* = 27) compared to the control (*p* < 0.001: ***) (Figure 3A). The additional incubation of IFN-β pre-treated M5 co-cultures with 10 ng/mL doxepin for 2 h led to an overall increase in the total cell number by 23.7% (*n* = 27) compared to the control (*p* ≤ 0.0001: ****). Moreover, incubation of IFN-β and doxepin led to a significant increase in the cell number by 17% compared to incubation with doxepin alone (*n* = 27) (*p* < 0.001: ***). However, incubation of the M30 co-cultures with 2000 U/mL IFN-β for 22 h led to a significant reduction in the total cell number by 15% compared to the control (*n* = 27) (*p* < 0.05: *) (Figure 3A). In contrast, the additional incubation with 10 ng/mL amitriptyline led to a significant increase in the cell number by 16.9% (*n* = 27) compared to the incubation with IFN-β alone (*p* < 0.05: *) (Figure 3A). Incubation with 10 ng/mL amitriptyline, 10 ng/mL doxepin or 2000 U/mL IFN-β caused no significant changes in the number of microglia detected by immunocytochemistry in the physiological M5 co-cultures (Figure 3B). In the pathological M30 co-cultures, pre-incubation with IFN-β and subsequent incubation with amitriptyline or doxepin led to a significantly increased number of microglia compared to incubation with amitriptyline, doxepin or IFN-β alone (*n* = 27) (*p* < 0.05: *) (Figure 3B).

### 3.3. Effects of Doxepin, Amitriptyline and IFN-β on Microglial Activation under Physiological and Pathological Conditions

Incubation of the physiological M5 co-cultures with 10 ng/mL amitriptyline or 10 ng/mL doxepin for 2 h, or with 2000 U/mL IFN-β for 22 h, showed a significant increase in the resting ramified type of microglia (RRT) (amitriptyline *p* < 0.01: **; doxepin *p* ≤ 0.0001: ****; IFN-β *p* < 0.05: *) (Figure 4A) and, in parallel, a significant decrease in the activated phenotype (RPT) (amitriptyline/doxepin/IFN-β *p* < 0.001: ***) (Figure 4B) by immunocytochemistry compared to the control. Pre-incubation with IFN-β and subsequent amitriptyline or doxepin addition also led to a significant decrease in activated microglia compared to incubation with one of the drugs alone under physiological M5 conditions (amitriptyline/IFN-β versus amitriptyline + IFN-β *p* < 0.01: **; doxepin/IFN-β versus doxepin + IFN-β *p* < 0.05: *) (*n* = 27). Treatment with amitriptyline, doxepin or IFN-β alone led to a highly significant increase in the resting ramified type of microglia (RRT) (Figure 4A) with a parallel, significant decrease in the activated phenotype (RPT) (Figure 4B) compared to the control under pathological, inflammatory M30 conditions (*p* ≤ 0.0001: ****) (*n* = 27). Treatment of IFN-β pre-incubated pathological M30 co-cultures with amitriptyline or doxepin significantly increased the amount of resting ramified microglia (Figure 4A) and, in parallel, reduced the microglial activation (Figure 4B) in a highly significant manner compared to the control (*n* = 27) (*p* ≤ 0.0001: ****).

### 3.4. Effects of Doxepin, Amitriptyline and IFN-β on Pro-Inflammatory Cytokines TNF-α and IL-6 in M5 and M30 Astrocyte–Microglia Co-Cultures

The TNF-α and IL-6 cytokine levels in cell culture supernatants were quantified by ELISA. Incubation with 10 ng/mL amitriptyline or 10 ng/mL doxepin for 2 h, or with 2000 U/mL IFN-β for 22 h, did not influence the IL-6 concentration in M5 (*n* = 9) as well as M30 (*n* = 9) co-culture supernatants compared to controls (Figure 5A,A-1) (*p* = ns). No significant changes in TNF-α cytokine concentration, in comparison to the controls, were measured in the supernatants of M5 (*n* = 9) and M30 (*n* = 9) co-cultures after incubation with the drugs (Figure 5B,B-1) (*p* = ns). The TNF-α and IL-6 cytokine levels in M5 and M30 co-cultures (*n* = 9) were not affected also after pre-incubation with IFN-β and subsequent amitriptyline or doxepin addition (Figure 5) (*p* = ns).

### 3.5. Effects of Doxepin, Amitriptyline and IFN-β on NO Concentration

Incubation with 10 ng/mL amitriptyline or 10 ng/mL doxepin for 2 h, or with 2000 U/mL IFN-β for 22 h alone, did not result in significant changes in NO concentrations under physiological (Figure 6A) or pathological (Figure 6B) conditions (*n* = 9). Moreover, co-incubation with amitriptyline or doxepin of IFN-β pre-treated co-cultures did not lead to NO changes (*n* = 9) (Figure 6) (*p* = ns).

## 4. Discussion

In this experimental study, the glial cell viability was unchanged under all conditions. Under pathological, inflammatory conditions, incubation with amitriptyline, doxepin or IFN-β alone caused a significant reduction in activated microglia and a parallel increase in resting microglia. In addition, treatment of IFN-β pre-incubated co-cultures with amitriptyline or doxepin reduced the microglial activation and increased the amount of resting ramified microglia in a highly significant manner. However, the pro-inflammatory TNF-α and IL-6 cytokine levels were not affected.

Positive effects of antidepressants in the context of MS are known. It has been shown that amitriptyline modulates depressive-like behavior in a myelin oligodendrocyte glycoprotein experimental autoimmune encephalomyelitis (MOG-EAE) model of MS in mice [51]. Moreover, amitriptyline showed anti-inflammatory effects in humans and experimental animal models of acute inflammation, as well as inhibiting the chronic inflammatory response to biomaterial in mice [52]. Amitriptyline led to a significant decrease in the activated rounded phagocytic type of microglia in our physiological M5 and pathological M30 co-cultures, confirming the inhibition of microglial activation and partially anti-inflammatory effects. However, in a neuropathic pain model in rats, amitriptyline treatment revealed an increase in the protein level of the microglia activation marker IBA-1 [33]. Otherwise, the administration of doxepin did not change the IBA-1 protein level [33]. This is consistent with our findings that treatment with doxepin resulted in a reduction in microglial activation under physiological and pathological conditions. Interestingly, amitriptyline and doxepin did not change significantly the total number of cells or microglia compared to untreated controls in M5 and M30 glial co-cultures. Previous studies showed that the treatment of a rat mixed glial culture with amitriptyline and nortriptyline decreased the release of both IL-1β and TNF-α, but the IL-1β and TNF-α mRNA levels in the mixed cell cultures were not affected [35]. Moreover, amitriptyline inhibited the LPS-induced IL-1β release by microglial cultures [35]. It reduced also the IL-1β protein levels in the astroglial cell line C6 [36]. In contrast to these findings, in our glial co-culture model, the pro-inflammatory TNF-α and IL-6 cytokine levels were unchanged after incubation with amitriptyline or doxepin. The exact causes remain unclear. Further studies with additional methodological techniques are useful, because each technique has its drawbacks. Therefore, the current obtained cytokine results using only one assay should be interpreted with caution. Previous data about doxepin in this context are limited. There are findings indicating that doxepin binds to normal, intact astrocytes in primary cultures [32]. Doxepin pre-treatment inhibited LPS-induced inflammation of the C6 glioma cells [53]. Further, doxepin prevented stress-induced memory impairments and decreased TNF-α levels in the rat hippocampus, if administered 1 mg/kg intraperitoneally for 21 days, but not for 5 mg/kg or 10 mg/kg concentrations, pointing toward a concentration-dependent effect [54]. In conclusion, these findings support the potential anti-inflammatory role of amitriptyline and doxepin against neuroinflammatory-related diseases in the brain.

IFN-β is an efficient, first-line drug in MS treatment [55]. The lack of endogenous IFN-β through gene deletion led to an increase in microglia activation, cytokine production and inflammation in EAE [56]. However, severe depression may occur during the treatment of MS with IFN-β [57] and therapy with antidepressants may be necessary in these patients. Our study provides novel insights about the effects of doxepin or amitriptyline on IFN-β pre-treated glial cells. IFN-β significantly increased the total number of cells, with an even greater increase after co-incubation with doxepin compared to untreated controls, but without a change in the microglial number in our physiological M5 co-cultures, indicating protective effects with regard to the cell amount and additional anti-inflammatory properties. This effect could not be observed for amitriptyline under physiological conditions. In our pathological, inflammatory M30 co-cultures, IFN-β significantly reduced the total number of cells without changing the microglia cell number. Interestingly, the incubation with doxepin or amitriptyline of IFN-β pre-treated co-cultures led to an increase in the total microglial cell number compared to incubation with the drugs alone. Treatment with IFN-β significantly reduced the amount of the activated rounded phagocytic type of microglia under physiological and pathological conditions, confirming its anti-inflammatory effects. This effect was observed also after co-incubation with doxepin or amitriptyline. These findings could be explained by the strong interaction of IFN-β with microglia, e.g., the activation of multiple intracellular cascades [58], the increased phagocytosis of apoptotic inflammatory cells by microglia upon IFN-β treatment [59] and the type I IFN receptor on microglial cells, which has protective functions in EAE [15]. Further, IFN-β regulates anti-inflammatory IL-10 production via toll-like receptors in microglia [60]. In addition, IFN-β may be produced by astrocytes, macrophages and ramified as well as activated microglia in MS/EAE lesions [15]. IFN-β was reported to act as an anti- and pro-inflammatory substance in microglial cells depending on the context [60,61]. After the exogenous addition of IFN-β, antigen-presenting functions such as the expression of MHC class II and co-stimulatory molecules are downregulated, while the production of pro-inflammatory cytokines and molecules such as TNF, IL-1β, IL-6, NO is enhanced [61]. IFN-β produced by microglia or added exogenously can cause the activation of phagocytic activity to remove myelin debris by microglia, which indicates a downregulation of pro-inflammatory factors [15,61]. These findings highlight the role of microglia, on the one hand, as IFN-β-producing cells and, on the other hand, the impact of IFN-β on microglia in CNS autoimmunity [15,61]. Moreover, changes in antigen-presenting functions and pro-inflammatory cytokine levels may contribute to side effects such as depressive symptoms caused by IFN-β treatment [61]. These two-sided effects could underline the observed increase in total microglia cells upon co-incubation of IFN-β pre-treated co-cultures with both antidepressants and the parallel reduction in microglial activation in our experiments. The two antidepressants, doxepin and amitriptyline, showed anti-inflammatory effects in our astrocyte–microglia co-culture model of inflammation independently of IFN-β. The key point of our study is that the both antidepressants did not interfere with the anti-inflammatory effect of IFN-β on glial cells; on the contrary, they show additional anti-inflammatory effects themselves. The co-incubation of IFN-β pre-treated glial co-cultures with doxepin or amitriptyline was not reported previously and offers new perspectives in the treatment of IFN-β side effects such as accompanying depression in MS.

In vivo studies with animal models showed that antidepressants can reduce inflammation and oxidative stress, e.g., inhibitory effects on the expression of inflammatory mediators, including cytokines, as well as on microgliosis and astrogliosis, were detected [62,63]. Some antidepressants act through the same inflammatory and oxidative stress mechanisms commonly reported to be disrupted in depression, and this may represent one of the mechanisms through which they exert additional antidepressive effects. Because microglia are involved in the regulation of inflammation and oxidative stress, and antidepressants can prevent microglial activation in vitro and in animal models, the inhibition of microglial activation may play an important role, contributing to antidepressive effects [62,64]. Moreover, the inhibition of pro-inflammatory cytokines by antidepressants can contribute to antidepressive behavior, e.g., doxepin prevented stress-induced memory impairments and decreased TNF-α levels in the rat hippocampus, explaining one possible mechanism of pharmacological symptom control by antidepressants [54]. However, there is only a very limited number of animal studies investigating the microglial and cytokine effects of antidepressants at the same time, e.g., the antidepressant fluoxetine did not affect the microglial activation marker IBA-1, but significantly reversed the increase in IL-1β levels induced by lipopolysaccharide and decreased the TNF-α levels, associated with behavioral changes in an animal model of major depression [62,65]. In sum, the behavioral and neurochemical alterations were reversed by fluoxetine [65]. These findings suggest the relationship between depressive-like symptoms in animal models induced by inflammation and partially anti-inflammatory effects of antidepressants, contributing to antidepressive mechanisms of action. However, it remains unclear whether the differential effects shown in our study (decrease in microglial activation and no change in cytokine levels) could interfere with the possible antidepressant effects of amitriptyline and doxepin at the behavioral level. Animal studies in this direction are necessary.

NO is involved as a key player in inflammation; however, both pro-inflammatory and anti-inflammatory actions are considered. There is evidence that NO plays a critical role also in the development of inflammatory demyelination, and microglia are involved in NO production in the CNS [66]. The link between oligodendrocyte death by ameboid microglial cells and NO production suggests that NO may play a role in the death of the oligodendrocyte and lesion formation in MS [66]. Moreover, it has been shown that NO is implicated in the pathogenesis of depression, e.g., NO production was increased in patients with depression [67]. However, the NO concentration was not changed after incubation with amitriptyline, doxepin or IFN-β or co-incubation with them under all conditions in our co-culture model. The anti-inflammatory effects of amitriptyline were related to a decrease in NO production, but data in microglial cultures were not reported previously [68]. Reduced IL-1β/TNF-α serum levels could be also implicated in the reduced NO production upon treatment with amitriptyline [68]. There are no previous studies available about doxepin’s effects on NO production. In addition, IFN-β contributed to a significant and dose-dependent increase in the NO concentration in microglial cultures [61]. However, in our study, we investigated only one concentration of IFN-β, so dose-dependent effects on NO could not be observed. This could explain why IFN-β treatment did not affect the NO concentration in our glial co-cultures.

In our astrocyte–microglia co-culture model, incubation with IFN-β, amitriptyline and doxepin has not shown effects on glial cell viability, indicating no cytotoxic effects. Data on amitriptyline are consistent with previous findings that amitriptyline-induced glial cell-line-derived neurotrophic factor production in astrocytes through the Gα (i/o) pathway supports cell plasticity, growth and survival [69]. Effects of doxepin on glial cells in this context were not reported previously. There is evidence that IFN-β has a bimodal effect on astrocytes depending on the regulation of the nuclear factor-kappa B (NF-κB) and depending on the dose of IFN-β [70]. IFN-β treatment may cause either the death of neighboring astrocytes due to exposure to high concentrations of the cytokine or promote astrocyte proliferation and survival for more distant astrocytes due to protection against cell death by low concentrations of the cytokine [70]. Further anti-proliferative effects of IFN-β were reported in glioma cell lines [71]. The fact that IFN-β did not influence the glial cell viability in our study could be due to the used concentration (2000 U/mL) and possible regulatory effects of our co-culture model.

A limitation of our study is the lack of data on cell connectivity, indicating novel findings on whole network effects with regard to anti-inflammation and therapy with antidepressants in in vitro and in vivo models. In contrast to the inhibition of microglial activation by amitriptyline and doxepin in interferon-β pre-treated co-cultures, the pro-inflammatory TNF-α and IL-6 cytokine levels were not affected in our study; therefore, additional investigations of cytokines including cytokine receptors (e.g., by high-sensitivity immunofluorescence, flow cytometry or at mRNA levels using reverse transcription polymerase chain reaction (RT-PCR)) are necessary to confirm the results in this regard. Other cytokines (e.g., IL-1β, TGF-β) and inflammation markers also need to be measured. Further, only one concentration of the drugs was used, so concentration-dependent effects could not be observed.

Even if doxepin and amitriptyline are not considered as a first choice for the treatment of depression, they offer further valuable effects concerning symptoms of insomnia, neuropathic pain, chronic headache and spastic pain in combination with sleep and anxiety disorders, which can occur in patients with MS together with additional symptoms of depression [28,29,30,31,72]. The advantage of the old antidepressants is that they are not selective, as with the modern serotonin re-uptake inhibitors, but have more effects. Therefore, we studied first the effects of doxepin and amitriptyline in our astrocyte–microglia co-culture model of inflammation. In future studies, it is useful also to investigate the effects of selective serotonin re-uptake inhibitors, which have mainly antidepressive effects.

## 5. Conclusions

In conclusion, the present study provides new insights into the endogenous inflammatory reaction and cytokine expression after incubation with interferon-β (IFN-β), amitriptyline and doxepin in physiological and pathological set-ups of an astrocyte–microglia co-culture model of inflammation, leading to a better understanding of non-neuronal cells and their pathological role in CNS diseases and treatment. The key finding of our study is that the both antidepressants do not interfere with the anti-inflammatory effect of IFN-β on glial cells; on the contrary, they exert additional anti-inflammatory effects themselves. Under pathological, inflammatory conditions, treatment with amitriptyline, doxepin or IFN-β alone, or co-incubation of IFN-β pre-treated co-cultures with both antidepressants, reduced the extent of inflammation, with the inhibition of microglial activation in a highly significant manner, but the pro-inflammatory TNF-α and IL-6 cytokine levels were not affected by the used drug concentrations. In addition, incubation with IFN-β, amitriptyline and doxepin has not shown effects on glial cell viability, suggesting no cytotoxic effects. This is the first in vitro study offering new perspectives in IFN-β treatment and accompanying depression regarding astrocytes and microglia.

Because both antidepressants are useful in the treatment of MS accompanied by depression, insomnia, neuropathic pain, chronic headache and spastic pain in combination with sleep and anxiety disorders, our study provides important results that may help to optimize the treatment of patients with MS.

## Figures and Tables

**Figure 1 brainsci-13-00493-f001:**
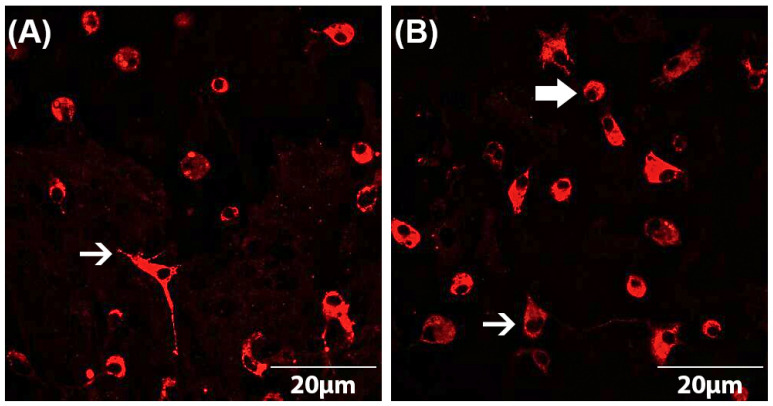
Microglia morphology in astrocyte–microglia co-cultures detected by immunocytochemistry. Staining with the antibody ED-1 (red) allowed the classification of microglia as resting ramified type (RRP) (white arrow in (**A**)), intermediate (INT) (thin, white arrow in (**B**)) and activated rounded phagocytic type (RPT) (thick, white arrow in (**B**)). Fluorescence microscopic images at a magnification of 600×.

**Figure 2 brainsci-13-00493-f002:**
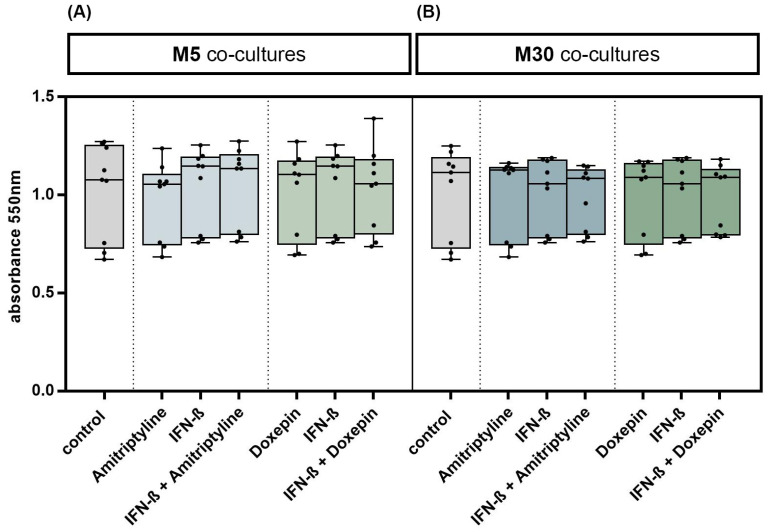
Glial cell viability was measured using MTT assay. Physiological (M5) (**A**) and pathological (M30) (**B**) astrocyte–microglia co-cultures were incubated with either 10 ng/mL amitriptyline or doxepin for 2 h, or with 2000 U/mL interferon-β (IFN-β) for 22 h. To investigate additional effects of antidepressants on IFN-β, cell cultures were pre-incubated with IFN-β (2000 U/mL) for 22 h and subsequently co-incubated with amitriptyline or doxepin for 2 h (*n* = 9). Untreated co-cultures were used as controls. The individual experimental conditions were compared by two tailed *t*-test. No significant differences between the individual conditions were detected (*p* = ns).

**Figure 3 brainsci-13-00493-f003:**
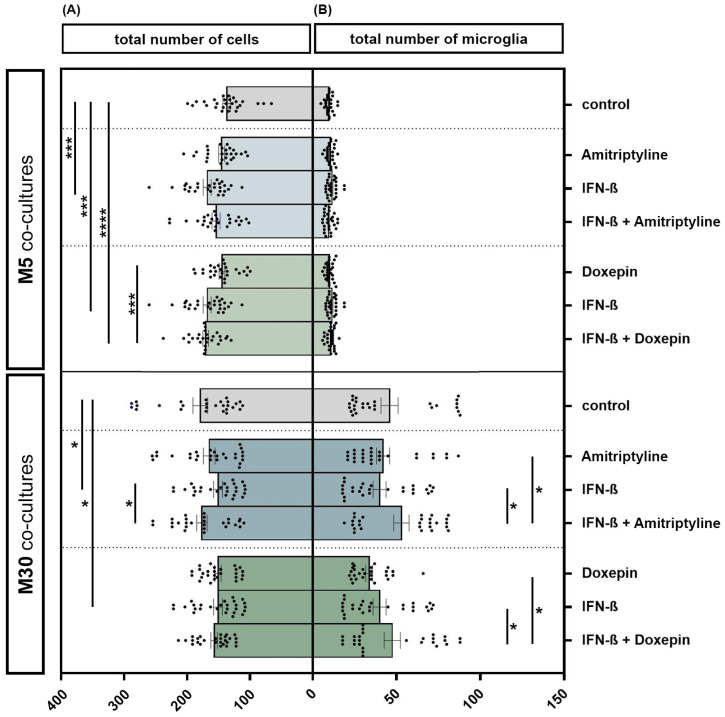
The total number of cells (**A**) and microglia (**B**) detected by immunocytochemistry in physiological (M5) and pathological (M30) co-cultures after incubation with 10 ng/mL amitriptyline, 10 ng/mL doxepin and 2000 U/mL interferon-β (IFN-β). In the physiological (M5) co-cultures, incubation with IFN-β for 22 h resulted in a significant increase in the cell number by 22.2% (*n* = 27) compared to the control (**A**). The additional incubation of IFN-β pre-treated M5 co-cultures with doxepin for 2 h led to an overall increase in the total cell number by 23.7% (*n* = 27) compared to the control. The direct comparison shows that the incubation of IFN-β and doxepin led to a significant increase in the cell number by 17% compared to incubation with doxepin alone (*n* = 27). In the M30 co-cultures, however, incubation with IFN-β led to a significant reduction in the total cell number by 15% compared to the control (*n* = 27). In contrast, the additional incubation with amitriptyline again led to a significant increase in the cell number by 16.9% (*n* = 27) compared to incubation with IFN-β alone (**A**). Incubation with 10 ng/mL amitriptyline, 10 ng/mL doxepin or 2000 U/mL IFN-β did not lead to significant changes in the number of microglia in the physiological M5 co-cultures (**B**). In the pathological M30 co-cultures, pre-incubation with IFN-β and subsequent incubation with amitriptyline or doxepin led to a significantly increased number of microglia compared to incubation with amitriptyline, doxepin or IFN-β alone (*n* = 27). Untreated co-cultures were used as controls. The individual experimental conditions were compared by two tailed *t*-test. Differences were considered significant at *p* < 0.05: *, *p* < 0.001: ***, *p* ≤ 0.0001: ****.

**Figure 4 brainsci-13-00493-f004:**
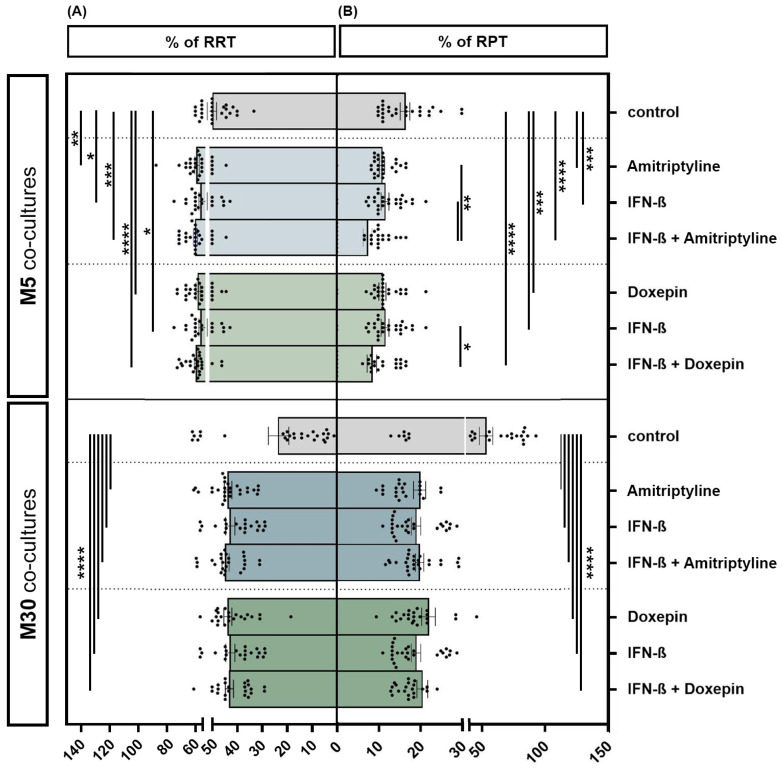
Microglial phenotypes after incubation with 10 ng/mL amitriptyline, 10 ng/mL doxepin and 2000 U/mL interferon-β (IFN-β) under physiological (M5) and pathological conditions (M30). Incubation of the physiological M5 co-cultures with amitriptyline, doxepin or IFN-β led to a significant increase in resting ramified type of microglia (RRT) (**A**) with a significant decrease in the activated phenotype (RPT) (**B**) compared to the control. Pre-incubation with IFN-β and subsequent amitriptyline or doxepin addition also resulted in a significant decrease in activated microglia compared to incubation with one of the drugs alone (*n* = 27). In the pathological M30 co-cultures, treatment with amitriptyline, doxepin or IFN-β alone led to a significant increase in resting ramified type of microglia (RRT) (**A**) with a parallel significant decrease in the activated phenotype (RPT) (**B**) compared to the control (*n* = 27). Treatment of IFN-β pre-incubated pathological M30 co-cultures with amitriptyline or doxepin resulted in a significant increase in resting ramified microglial type (RRT) (**A**) with a parallel significant decrease in the activated phenotype (RPT) (**B**) compared to the control (*n* = 27). Untreated co-cultures were used as controls. The individual experimental conditions were compared by two tailed *t*-tests. Differences were considered significant at *p* < 0.05: *, *p* < 0.01: **, *p* < 0.001: ***, *p* ≤ 0.0001: ****.

**Figure 5 brainsci-13-00493-f005:**
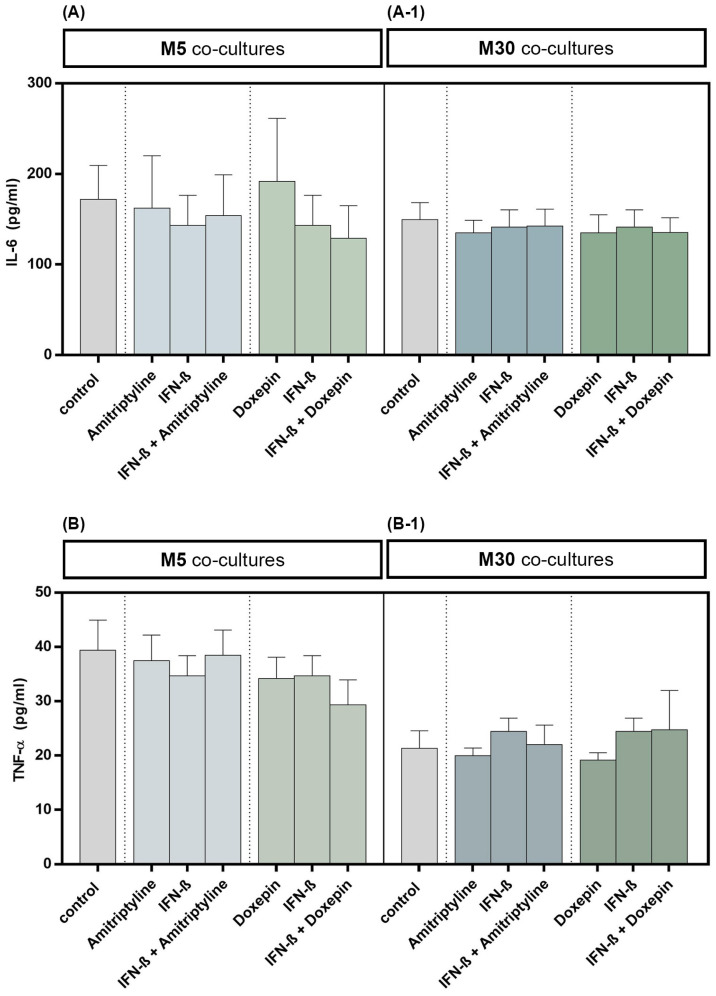
Interleukin-6 (IL-6) (**A**,**A-1**) and tumor necrosis factor-α (TNF-α) (**B**,**B-1**) cytokine concentrations quantified by ELISA in the supernatants of physiological (M5) (**A**,**B**) and pathological (M30) (**A-1**,**B-1**) co-cultures after incubation with 10 ng/mL amitriptyline or 10 ng/mL doxepin for 2 h, or with 2000 U/mL interferon-β (IFN-β) for 22 h. Incubation with the drugs did not influence the IL-6 levels in M5 (*n* = 9) as well as M30 (*n* = 9) co-culture supernatants compared to controls. No significant change in TNF-α cytokine levels, in comparison to the controls, was measured in the supernatants of M5 and M30 co-cultures after incubation with the drugs (*n* = 9). The Four Parameter Logistic Regression protocol from myassays.com (https://www.myassays.com; accessed on 1 December 2021) was used in the analysis and statistics of the ELISA results. Untreated co-cultures were used as controls. The individual experimental conditions were compared by two tailed *t*-test. No significant differences between the individual conditions were detected (*p* = ns).

**Figure 6 brainsci-13-00493-f006:**
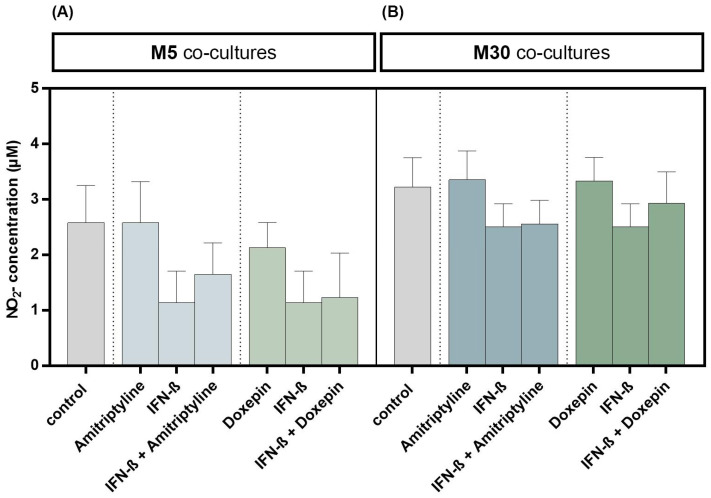
Nitric oxide (NO) concentration determined based on measurement of nitrite (NO_2_) as a primary breakdown product of NO using Griess (NO) assay in the physiological (M5) (**A**) and pathological (M30) (**B**) astrocyte–microglia co-cultures after incubation with 10 ng/mL amitriptyline or 10 ng/mL doxepin for 2 h, or with 2000 U/mL interferon-β (IFN-β) for 22 h. Incubation with the antidepressants or IFN-β alone did not result in significant changes in NO_2_ concentrations in physiological (**A**) or pathological (**B**) co-cultures (*n* = 9). Moreover, the co-incubation with amitriptyline or doxepin of IFN-β pre-treated co-cultures did not lead to NO_2_ changes (*n* = 9). Untreated co-cultures were used as controls. The individual experimental conditions were compared by two-tailed *t*-test. No significant differences between the individual conditions were detected (*p* = ns).

## Data Availability

The datasets used or analyzed during this study are available from the corresponding author upon reasonable request.

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
