# Peer review of "Inhibition of Microglial Activation by Amitriptyline and Doxepin in Interferon-β Pre-Treated Astrocyte–Microglia Co-Culture Model of Inflammation†"

_brainsci, 2023, doi:10.3390/brainsci13030493_

Round 1

Reviewer 1 Report

The study aimed to gain insights into the endogenous inflammatory reactions and cytokine expressions of non-neuronal cells in physiological and pathological settings, leading to a better understanding of their role in central nervous system diseases and treatment. The key finding of the study is that both antidepressants, amitriptyline and doxepin, did not interfere with the anti-inflammatory effect of IFN-β on glial cells but instead exerted additional anti-inflammatory effects themselves. Treatment with these drugs, either alone or in combination with IFN-β, reduced inflammation and inhibited microglial activation significantly under pathological, inflammatory conditions. However, the pro-inflammatory cytokine levels were not affected by the used drug concentrations. Furthermore, incubation with IFN-β, amitriptyline, and doxepin did not show any cytotoxic effects on glial cell viability. In my opinion, this article has a high standard of methodology and reporting because it provides important results that may help to optimize the treatment of patients with multiple sclerosis.

Author Response

The study aimed to gain insights into the endogenous inflammatory reactions and cytokine expressions of non-neuronal cells in physiological and pathological settings, leading to a better understanding of their role in central nervous system diseases and treatment. The key finding of the study is that both antidepressants, amitriptyline and doxepin, did not interfere with the anti-inflammatory effect of IFN-β on glial cells but instead exerted additional anti-inflammatory effects themselves. Treatment with these drugs, either alone or in combination with IFN-β, reduced inflammation and inhibited microglial activation significantly under pathological, inflammatory conditions. However, the pro-inflammatory cytokine levels were not affected by the used drug concentrations. Furthermore, incubation with IFN-β, amitriptyline, and doxepin did not show any cytotoxic effects on glial cell viability. In my opinion, this article has a high standard of methodology and reporting because it provides important results that may help to optimize the treatment of patients with multiple sclerosis.

REPLY: We thank the Reviewer for the comment.

Reviewer 2 Report

Dear authors, it has been a pleasure to review this manuscript. I found it really interesting as it brings novel aspects to the study of depression in patients with Multiple Sclerosis.

What direct implication do the authors think the results obtained in this study may have?

Author Response

Dear authors, it has been a pleasure to review this manuscript. I found it really interesting as it brings novel aspects to the study of depression in patients with Multiple Sclerosis.

What direct implication do the authors think the results obtained in this study may have?

REPLY: The present study provides new insights into endogenous inflammatory reaction and cytokine expression after incubation with interferon-β (IFN-β), amitriptyline and doxepin in physiological and pathological set-ups of an astrocyte-microglia co-culture model of inflammation, leading to better understanding of non-neuronal cells and their pathological role in CNS diseases and treatment. The key finding of our study is that the both antidepressants do not interfere with the anti-inflammatory effect of IFN-β on glial cells, on the contrary, they exert additional anti-inflammatory effects themselves. Because both antidepressants are useful in the treatment of MS accompanied by depression, insomnia, neuopathic pain, chronic headache, spastic pain in combination with sleep and anxiety disorders, our study provides important results that may help to optimize the treatment of patients with MS.

These statements were included in our manuscript (page 15-16, line 552-570).

Reviewer 3 Report

  • The aim of the paper is to discuss results of co-incubation of IFN-β pre-treated co-cultures with chosen antidepressants on the extent of inflammation with microglial activation, as well as the concentration of TNF-α and IL-6 levels. In the introduction, I miss the hypothesis statement at the end of the section. Generally, the manuscript is clear, relevant for the field and presented in a well-structured manner. The cited references are of different newness, however, they are relevant to the field. The experimental design is appropriate to answer the question form the introduction section. The details given in the methods section are far enough. Overall, the presented research is a valuable addition to the existing knowledge. However, amitriptiline and doxepin are out-fashioned antidepressants and are not used  often any more. The study would have better practical implications testing incubation with modern antidepressants. I would discuss this issue in the limitations paragraph.

Author Response

The aim of the paper is to discuss results of co-incubation of IFN-β pre-treated co-cultures with chosen antidepressants on the extent of inflammation with microglial activation, as well as the concentration of TNF-α and IL-6 levels. In the introduction, I miss the hypothesis statement at the end of the section. Generally, the manuscript is clear, relevant for the field and presented in a well-structured manner. The cited references are of different newness, however, they are relevant to the field. The experimental design is appropriate to answer the question form the introduction section. The details given in the methods section are far enough. Overall, the presented research is a valuable addition to the existing knowledge. However, amitriptiline and doxepin are out-fashioned antidepressants and are not used  often any more. The study would have better practical implications testing incubation with modern antidepressants. I would discuss this issue in the limitations paragraph.

REPLY: We thank the Reviewer for the advice.

We hypothesized that amitriptyline and doxepin did not attenuate the anti-inflammatory effect of IFN-β treatment with regard to glia-mediated inflammation. (page 3, line 130-132)

Even if doxepin and amitriptyline are not considered as first choice for the treatment of depression, they offer further valuable effects concerning symptoms of insomnia, neuopathic pain, chronic headache, spastic pain in combination with sleep and anxiety disorders, which can occur in patients with MS together with additional symptoms of depression. The advantage of the old antidepressants is that they are not selective like the modern serotonin re-uptake inhibitors, but have more effects. Therefore, we studied first the effects of doxepin and amitriptyline in our astrocyte-microglia co-culture model of inflammation. In future studies, it is useful also to investigate the effects of selective serotonin re-uptake inhibitors, which have mainly anti-depressive effects.

We have included this statement in our limitations. (page 15, line 542-550)

Reviewer 4 Report

Review

1. Page 1, line 27: What does mean “MTT”?

2. Page 2, line 49: “small cell body and a high ramification”. Please describe the morphology of the activated microglia.

3. Page 2, line 70: “Patients with history of depression have a higher risk to develop depression under IFN-β treatment”. What is the biological mechanism that has been implicated in the increased risk of depression in patients receiving IFN-β treatment?

4. Page 2, line 87: “are doxepin and amitriptyline”. Regularly, these types of drugs are not the first choice in the treatment of depression, and they also have serious side effects (for example, cardiovascular problems). Authors should justify the use of these two drugs in their study, and not that of other antidepressants that are usually first choice, such as selective serotonin reuptake inhibitors.

5. Page 11, line 361: “The NO concentrations were determined by Griess (NO) assay”. Information repeated with the methodology. Delete please.

6. Page 12, line 352: “confirming inhibition of microglial activation and antiinflammatory effects” – “confirming inhibition of microglial activation and partially anti-inflammatory effects”. The foregoing considering that no changes were observed in the levels of proinflammatory cytokines, nor in NO.

7. Page 12, lines 414-416: Considering the differential effects shown in this study (decrease in microglial activation and no change in cytokine levels), authors should suggest whether this could interfere with possible antidepressant effects at the behavioral level. Please, support with some other study where these differential effects have happened in some animal model, and the antidepressant effect has been seen.

Author Response

  1. Page 1, line 27: What does mean “MTT”?

REPLY: MTT means 3-(4,5-dimethylthiazol-2-yl)-2,5-diphenyltetrazolium bromide. We have included this in the sentence.

  1. Page 2, line 49: “small cell body and a high ramification”. Please describe the morphology of the activated microglia.

REPLY: We have added the following statement: “Activated microglia are characterized by a large cellular diameter, rare short processes and several cytoplasmic vacuoles”. (page 2, line 52-53)

  1. Page 2, line 70: “Patients with history of depression have a higher risk to develop depression under IFN-β treatment”. What is the biological mechanism that has been implicated in the increased risk of depression in patients receiving IFN-β treatment?

REPLY: Depression was discussed to occur under treatment with IFN-β. Further, a history of depression was discussed to be a risk factor to develop depression under IFN-β treatment. The interaction between immune, endocrine and neuronal pathways is assumed to contribute to IFN-related depression. Possible biochemical/biological mechanisms include: (1) IFN-therapy (α, β) induced hyperactivity by the hypothalamo-pituitary-adrenal axis, releasing corticotrophin releasing hormone (CRH), which increases adrenocorticotropic hormone and hence adrenal corticosterone release, and decreases serotonin and noradrenaline; (2) modulation of mood and sleep behavior by activation of pro-inflammatory cytokines by IFN-α (e.g., interleukin-6 (IL-6), interleukin-1 (IL-1), tumor necrosis factor-α (TNF-α), caspase-4 and caspase-8); (3) µ-opioid receptor activation by IFN (α, β), leading to an increase in brain prostaglandin E2 and different regulation of enzymes (e.g., indolamine 2,3-dioxygenase (IDO), kynureninase),  influencing the N-methyl-D-aspartate (NMDA)-receptor, which results finally in excitotoxicity. (page 2, line 74-84)

  1. Page 2, line 87: “are doxepin and amitriptyline”. Regularly, these types of drugs are not the first choice in the treatment of depression, and they also have serious side effects (for example, cardiovascular problems). Authors should justify the use of these two drugs in their study, and not that of other antidepressants that are usually first choice, such as selective serotonin reuptake inhibitors.

REPLY: Even if doxepin and amitriptyline are not considered as first choice for the treatment of depression, they offer further valuable effects concerning symptoms of insomnia, neuopathic pain, chronic headache, spastic pain in combination with sleep and anxiety disorders, which can occur in patients with MS together with additional symptoms of depression. The advantage of the old antidepressants is that they are not selective like the modern serotonin re-uptake inhibitors, but have more effects. Therefore, we studied first the effects of doxepin and amitriptyline in our astrocyte-microglia co-culture model of inflammation. In future studies, it is useful also to investigate the effects of selective serotonin re-uptake inhibitors, which have mainly anti-depressive effects.

We have included this statement in our limitations. (page 15, line 542-550)

  1. Page 11, line 361: “The NO concentrations were determined by Griess (NO) assay”. Information repeated with the methodology. Delete please.

REPLY: We have deleted the repeated information.

  1. Page 12, line 352: “confirming inhibition of microglial activation and antiinflammatory effects” – “confirming inhibition of microglial activation and partially anti-inflammatory effects”. The foregoing considering that no changes were observed in the levels of proinflammatory cytokines, nor in NO.

REPLY: We have changed the sentence. (page 13, line 406)

  1. Page 12, lines 414-416: Considering the differential effects shown in this study (decrease in microglial activation and no change in cytokine levels), authors should suggest whether this could interfere with possible antidepressant effects at the behavioral level. Please, support with some other study where these differential effects have happened in some animal model, and the antidepressant effect has been seen.

REPLY: In vivo studies with animal models showed that antidepressants can reduce inflammation and oxidative stress e.g., inhibitory effects on the expression of inflammatory mediators, including cytokines, as well as on microgliosis and astrogliosis were detected. Some antidepressants act through the same inflammatory and oxidative stress mechanisms commonly reported to be disrupted in depression, and this may represent one of the mechanisms through which they exert additional anti-depressive effects. Because microglia are involved in the regulation of inflammation and oxidative stress, and antidepressants can prevent microglial activation in vitro and in animal models, inhibition of microglial activation may play an important role, contributing to anti-depressive effects. Also inhibition of pro-inflammatory cytokines by antidepressants can contribute to anti-depressive behavior e.g., doxepin prevented stress-induced memory-impairments and decreased TNF-α levels in the rat hippocampus, explaining one possible mechanism of pharmacological symptom control by antidepressants. However, there is only a very limited number of animal studies, investigating the microglial and cytokine effects of antidepressants at the same time e.g., the antidepressant fluoxetine did not affect the microglial activation marker IBA-1, but significantly reversed the increase in IL-1β levels induced by lipopolysaccharide and decreased the TNF-α levels, associated with behavioral changes in an animal model of major depression. In sum, the behavioral and neurochemical alterations were reversed by fluoxetine. These findings suggest the relationship between depressive-like symptoms in animal models induced by inflammation and partially anti-inflammatory effects of antidepressants, contributing to anti-depressive mechanisms of action. However, it remains unclear whether the differential effects shown in our study (decrease in microglial activation and no change in cytokine levels) could interfere with possible antidepressant effects of amitriptyline and doxepin at the behavioral level. Animal studies in this direction are necessary.

We have included these statements in our discussion. (page 14, line 474-498)